# Amphiphilic Pentablock Copolymers Prepared from Pluronic and ε-Caprolactone by Enzymatic Ring Opening Polymerization

**DOI:** 10.3390/ijms23031390

**Published:** 2022-01-26

**Authors:** Ahmed Abd El-Fattah, Elizabeth Grillo Fernandes, Federica Chiellini, Emo Chiellini

**Affiliations:** 1Laboratory of Bioactive Polymeric Materials for Biomedical and Environmental Applications (BIOlab), UdR INSTM, Department of Chemistry and Industrial Chemistry, University of Pisa, Via Vecchia Livornese 1291, San Piero a Grado, 56010 Pisa, Italy; federica.chiellini@unipi.it (F.C.); emo.chiellini@unipi.it (E.C.); 2Department of Materials Science, Institute of Graduate Studies and Research, Alexandria University, Alexandria 21526, Egypt; 3Department of Chemistry, College of Science, University of Bahrain, Sakhir P.O. Box 32038, Bahrain; 4Department of Metallurgical and Materials Engineering, Polytechnic School, University of São Paulo, São Paulo 05508-070, Brazil

**Keywords:** *Candida antarctica* lipase B enzyme, ring-opening polymerization, ε-caprolactone, Pluronic, morphology

## Abstract

Amphiphilic copolymers are appealing materials because of their interesting architecture and tunable properties. In view of their application in the biomedical field, the preparation of these materials should avoid the use of toxic compounds as catalysts. Therefore, enzymatic catalysis is a suitable alternative to common synthetic routes. Pentablock copolymers (CUC) were synthesized with high yields by ring-opening polymerization of ε-caprolactone (ε-CL) initiated by Pluronic (EPE) and catalyzed by *Candida antarctica* lipase B enzyme. The variables to study the structure–property relationship were EPEs’ molecular weight and molar ratios between ε-CL monomer and EPE macro-initiator (M/In). The obtained copolymers were chemically characterized, the molecular weight determined, and morphologies evaluated. The results suggest an interaction between the reaction time and M/In variables. There was a correlation between the differential scanning calorimetry data with those of X-ray diffraction (WAXD). The length of the central block of CUC copolymers may have an important role in the crystal formation. WAXD analyses indicated that a micro-phase separation takes place in all the prepared copolymers. Preliminary cytotoxicity experiments on the extracts of the polymer confirmed that these materials are nontoxic.

## 1. Introduction

The great interest in amphiphilic block copolymers for biomedical applications, regarding specifically drug delivery and tissue engineering, is due to their unique chain architecture and self-assembling capabilities. In fact, these highly ordered self-assembled structures display interesting phase behavior in aqueous media with propensity to yield core-shell-type nanoparticles or polymeric micelles with a hydrophobic inner core and a hydrophilic outer shell. Such particles are amenable to entrap drugs or bioactive agents and deliver them under controlled conditions to selected body compartments [1].

Polymeric micelles have good structural stability even at a fairly low critical micelle concentration (CMC) and display slow dissociation of the polymeric chains. These structures allow for a good control over the particle size and the hydrophobic drug solubilization, and are, thus, suitable for drug delivery applications [2,3,4].

Pluronics (EPE) are a class of flexible and biocompatible polyether, cleared by US FDA for use as food additives and pharmaceutical ingredients. These materials are amphiphilic ABA-type triblock copolymers based on ethylene oxide (EO) and propylene oxide (PO) arranged in the following structure: EO_x_-PO_y_-EO_x_. EPE copolymers have the capability of forming micelles in an aqueous environment depending on the hydrophilic/hydrophobic balance and the solution temperature [5]. Their (CMC) and critical micelle temperature (CMT) are, however, fairly high, due to the low hydrophobicity of the PO block at ambient temperature. In order to decrease the CMC and the CMT of the EPE copolymer, and thus its micelle stability, the hydrophobic content has to be increased. Thus, for example, hydrophobic poly(ε-caprolactone) (PCL) has been grown on EPE hydroxyl end groups, since it is a well-known biocompatible and biodegradable polymer, widely used in the biomedical field for controlled drug delivery [6,7].

Ring-opening polymerization (ROP) of ε-caprolactone (ε-CL) initiated by the EPE hydroxyl end-groups and catalyzed by tin 2-ethylhexanoate [Sn(Oct)_2_] has been, so far, the most common synthetic route for PCL–EPE–PCL pentablock copolymers. However, although FDA approved, tin cytotoxicity is a concern and its use should be avoided, especially when materials are to be used in vivo [8,9,10].

The use of enzymatic catalysis for the preparation of materials for biomedical applications allows circumvention of the toxicity issues related to the commonly used metal catalysts. Moreover, enzymes provide high catalytic activity, large acceleration of the reaction rate under mild conditions, and high selectivity of the substrates [11]. *Candida antarctica* lipase B (CALB) is an enzyme that catalyzes the hydrolysis of fatty acid esters in an aqueous environment; it can also be stable in organic solvents and used as a catalyst for esterification and transesterification reactions. Since 1993, polymerizations of various cyclic compounds using lipase catalysis have been performed and lactones were the most extensively investigated to prepare aliphatic polyesters, thus providing a valid alternative to metal catalysts for the synthesis of polyesters via ROP [12,13].

In the present work, the synthesis of polyester–polyether pentablock copolymers based on the ROP of ε-CL, as initiated by dihydroxyl terminated EPE and catalyzed by CALB, was performed. Different monomer-to-initiator ratios (M/In) were used in order to investigate the structure–property relationships of the obtained materials. The prepared polymers were characterized by ^1^H-NMR, FT-IR, SEC, thermal analysis, WAXD, and optical microscopy. Preliminary cytotoxicity evaluation of the materials was assessed.

## 2. Results and Discussion

PCL–EPE–PCL (CUC) pentablock copolymers were synthesized by the reaction of dihydroxyl-terminated EPE macro-initiator with the enzyme-activated ε-CL (Figure 1).

The polymers were recovered with yields of 70–85%, which is in accordance with the yield value of 86% found in the homopolymerization of ε-CL at the same reaction condition [14].

### 2.1. Size Exclusion Chromatography (SEC)

SEC traces of the copolymers showed monomodal peaks, as shown in Figure 1, which indicates that CUC copolymers were produced without homopolymerization of PCL. The values of molecular weight (${\bar{M}}_{n_{SEC}}$) and dispersity (Đ) are listed in Table 1.

The narrow Đ confirms that the polymerization was fairly controlled. The mean values and standard deviation for the CUC1–CUC4 and CUC5–CUC8 series were 1.66 ± 0.11 and 1.51 ± 0.03, respectively.

### 2.2. Molecular Characterization

Figure 2 shows typical FT–IR spectra of the CUC1–CUC4 copolymer series (see Table 1) compared with pristine Pluronics F38 (EPE(F38)) and a PCL homopolymer. CUC pentablock copolymers exhibit peaks characteristic of both EPE and PCL. The absorption band at 1107 cm^−1^ is attributed to the characteristic C–O–C stretching vibrations of the ether units of EPE, while the peak at 1243 cm^−1^ corresponds to the absorption of –(C=O)–O–C– bond stretching vibrations of PCL units. The strong sharp band at 1725 cm^−1^ is attributed to the stretching vibration of the ester carbonyl group of PCL block. It is worth noting that, as expected, the intensity of the ester carbonyl group increases with increasing content of PCL in the copolymers. Liu et al. found similar results in the PCL–EPE(L35)–PCL copolymers synthesized with stannous octoate as a catalyzer [15].

^1^H-NMR spectroscopy provides information about both the chemical composition and the number-average molecular weight (${\bar{M}}_{n_{CUC}}$) of the copolymers. The ^1^H-NMR spectrum of CUC4 copolymer is shown in Figure 3, as a typical example. The sharp singlet at 3.63 ppm is attributed to methylene protons of PEO blocks (–OCH_2_CH_2_–) in EPE. The two small peaks at 3.4 and 3.5 ppm are attributed to –CH– and –CH_2_– groups, respectively, while the doublet peak at 1.15 ppm belongs to –CH_3_ of PPO block in EPE. The spectrum exhibits two equally intense triplets at 4.05 and 2.30 ppm assigned to –CH_2_OOC– and –OCCH_2_C–, respectively, as well as two multiplets at 1.4 and 1.6 ppm, which are assigned to methylene protons –(CH_2_)_3_ in PCL blocks. Although the characteristic peak of the terminal –CH_2_OH group was expected at around 3.5 ppm, it could not be observed due to its small intensity and the overlapping with the peaks of –CH_2_CH– in PPO block at ca. 3.4 and ca. 3.5 ppm. Ha et al. obtained results that agree with those of the present work [16].

The degree of polymerization (*DP_PCL_*) and the number-average molecular weight of PCL segments (${\bar{M}}_{n_{PCL}}$) can be calculated from the intensity ratio of the methyl protons at 1.15 ppm of PPO in EPE and the methylene protons at 4.05 ppm of the PCL blocks by using Equation (1):(1)$$DP_{PCL}=DP_{PPO}\frac{3\times I_{4.05}}{2\times I_{1.05}}$$
where, *DP_PPO_* is the degree of polymerization of PPO in EPE, *I*_4.05_ is the peak intensity of CH_2_ group in CL block, and *I*_1.15_ is the peak intensity of CH_3_ group of PO in EPE.

Considering that the CUC chain has one PCL block at each end of EPE, the calculated value of *DP_PCL_* should be divided by 2. Values of ${\bar{M}}_{n_{CUC}}$ of copolymers and ${\bar{M}}_{n_{theor}}$ can be computed according to Equations (2) and (3), respectively:(2)$${\bar{M}}_{n_{CUC}}={\bar{M}}_{n_{EPE}}+DP_{PCL}\times 114$$
(3)$${\bar{M}}_{n_{theor}}={\bar{M}}_{n_{EPE}}+\left(M/In\right)\times 114$$
where, 114 is the molecular weight of ε-CL monomeric unit, and ${\bar{M}}_{n_{EPE}}$ and ${\bar{M}}_{n_{theor}}$ are number-average molecular weight of EPE macro-initiator and theoretical molecular weight of copolymers, respectively.

The degree of polymerization of PCL (*DP_PCL_*) as well as the molecular weight of the copolymers (${\bar{M}}_{n_{CUC}}$) increased with increasing molar feed ratio of monomer to EPE (*M/In*) (Table 1). However, the calculated *DP_PCL_* values from ^1^H-NMR data somehow disagree with the theoretical ones (*M*/*In*), except for the first two of each series (CUC1 and CUC5). The *DP_PCL_* values in the series with EPE(F38) (CUC1–CUC4) were about 30% lower than M/In. On the other hand, in the series with EPE(F68) (CUC5–CUC8), the difference between calculated and theoretical values were between 12% and 25%, increasing with the increase in *M/In* rate. Zhu et al. found the same dependence between *DP_M_* and *M/In* in the polymerization of diacrylated EPE(F127)/oligoester copolymers catalyzed with stannous octoate [17]. Comparable behavior was found in the values of dispersity (Đ). That is, the values of Đ are slightly higher for larger differences between calculated and theoretical *DP_PCL_*. As the reaction time was set at 4 h, it is possible that it was not sufficient to convert all ε-CL monomer and that the choice of this time depends of *M/In* ratio.

The “quasi-living” or controlled characteristic of the enzymatic polymerization is suggested in Figure 4. The NMR molecular weights changed linearly, with *DP_PCL_* overlapping the theoretical trace (${\bar{M}}_{n_{theor}}$ × *M/In*) for both series of CUC copolymers, even showing a gap between *DP_PCL_* and *M*/*In*. On the other hand, this relationship presented different behavior in the SEC molecular weights and probably depends on the values of the EPE used as the macro-initiator. The ${\bar{M}}_{n_{SEC}}$ values of EPE(F38) series (${\bar{M}}_{n_{EPE}}$ = 4700) were lower than those of the theoretical trace for M/In higher than 105 (*DP_PCL_* > 74), while ${\bar{M}}_{n_{SEC}}$ values of EPE(F68) series (${\bar{M}}_{n_{EPE}}$ = 8400) were slightly higher than the theoretical ones and exhibited low linearity. It should be remarked that NMR spectroscopy is an absolute method for molecular weight determination, while the SEC method is relative and depends not only on calibration standard, but also on the hydrodynamic volume of the polymer in the specific solvent used in the analysis. Therefore, it is possible that the molecular weight calibration based on PS standards may not give accurate ${\bar{M}}_{n_{SEC}}$ for CUC copolymers due to a compromise between chain length and polymer architecture. The narrow polydispersity indices, varying from 1.47 to 1.76 (Table 1), confirm that the polymerization was fairly controlled.

### 2.3. Solid-State Characterization

#### 2.3.1. Differential Calorimetry Scanning (DSC)

The correlation between the thermodynamic parameters of the pentablock copolymers and their composition was evaluated by DSC. CUC1–CUC4 series represent the typical DSC traces of CUC copolymers, which are compared with the traces of pristine EPE(F38) and of a reference PCL homopolymer in Figure 5.

In the first heating scan, EPE(F38) and PCL presented only one melting temperature peak (T_m_) at 62 °C and 68 °C, respectively (Table 2). The temperature of these transitions decreased by 4 °C and 8 °C in the second heating scan, respectively. Besides, EPE(F38) melting transition presented a shoulder at the lower temperature side. The appearance of a shoulder in the melting traces of EPE triblock copolymers was proposed to correspond to the lamellar structure with different thickness, based on observations in small-angle X-ray scattering (SAXS) [18]. It was suggested that the origin of these multi-lamellar structures is the PEO block dispersity length or the presence of di-block copolymers that can be found in the commercial products. This could explain the low crystallization temperature (T_c1_) of EPE(F38) at –8 °C observed in the cooling at 10 °C/min, other than the principal one (T_c2_) at 30 °C (Table 2). Another consideration is that the profiles of the first heating scan represent the thermal characteristics of powder samples that were obtained; this means that the crystalline organization originated from solution-precipitation treatment. On the other hand, melting transitions observed in the second heating scan have a thermal history corresponding to melt-cooling at 10 °C/min. A similar dependence of the crystal organization with the thermal history has been found in the literature [18,19]. The supercoolings (ΔT = T_m_−T_c_) between T_m_ of the second heating and T_c_ for both EPE(F38) and PCL polymers were of 28 °C (= T_m2_−T_c2_) and 25 °C (= T_m3_−T_c3_), respectively, under the experimental conditions. Considering the series with the EPE(F68) (CUC5–CUC8), it is observed that its ΔT value was equivalent to the previous series, that is, 27 °C. Regarding the two series of pentablocks, more than two crystallization peaks were not always observed. Apparently, the size of the EPE segment in the pentablock copolymer influences the behavior of its DT with the increase in the PCL segment. Thus, the results suggest that the effect of increasing the molecular mass of the PCL segment is to increase the DT of the pentablock copolymer, bringing consequences to its morphology [18,20].

Both CUC copolymer series presented up to three melting peaks that are overlapped and/or a shoulder due to the proximity of the EPE and PCL T_ms_; these characteristics have been found in the literature for equivalent copolymers [21,22]. Besides, DSC profiles are very comparable on both first and second heating scan, with practically the same T_ms_ (Table 2 and Table 3). This means that crystal organization in the CUC copolymers is stable and probably independent of thermal history.

In the CUC1–CUC4 series, the peak area and the T_m1_ of the first peak (Figure 5) decreases with relative decreasing of EPE block length, until it becomes almost imperceptible in the CUC4 copolymer, as indicated by the arrows (Figure 5). The same behavior was found by Bogdanov et al., who attributed this transition to the ethylene glycol (EG) block in the CL–EG–CL triblock copolymers [22]. Furthermore, the authors concluded that the PCL block crystallized before it would be influencing the PEG crystallization process. Considering the T_m1_ values of the CUC1–CUC4 series, it is verified that they changed from 44 °C to 28 °C with the increase in the PCL segment, going from CUC1 to CUC3, respectively, due to the effect of PCL crystallization [22]. On the other hand, the second peak with T_m2_ values over 57 °C (CUC1) shifts to a higher temperature overlapping the third peak in copolymers with higher *M/In* ratios (Table 1—CUC3 and CUC4). The values of T_m3_ of CUC copolymers ranging from 61 °C to 69 °C showed a linear relationship with the wt% of the PCL, as represented by Equation (4), which indicates that the melting temperatures are controlled by its molecular weight (or its block size).
(4)$$T_{m3}=40.243+0.445\times \mathrm{PCL}\;wt\%$$

The first peak at the highest crystallization temperature is attributed to PCL, followed by that related to the ethylene oxide (EO) segment of EPE. So, it was previously observed that the crystallization of EPE(F38) presents two peaks. Accordingly, in the present work, T_c1_ and T_c2_ correspond to crystallization of EO block and T_c3_ to CL block. All crystallization temperature values decreased with increasing of the CL block length. However, this behavior was more marked for T_c1_. Similar thermal behavior was found in the CUC5–CUC8 series where the temperatures of the first melting peak (T_m1_) decrease and those of the third (T_m3_) increase with increasing of PCL/PEO ratios. However, the changes in T_ms_ with compositions were less marked in the CUC5–CUC8 series and the linear dependence of T_m3_ on the wt% of the PCL is given by Equation (5). Besides, in the cooling scan only two crystallizations were detected.
(5)$$T_{m3}=57.086+0.143\times \mathrm{PCL}\;wt\%$$

The principal difference between the two CUC series is the macro-initiator molecular weights and their Đ, as shown in the Table 1. In the series with EPE(F38), the values of Đ were between 1.47 and 1.76, which are higher than those with EPE(F68) (1.48–1.56). Consequently, it can be supposed that not only the molecular weight (and, therefore, the length of the amorphous PO block and copolymer) determine the crystal organization, but probably its dispersion, as proposed by Zhang et al. [18].

The thermal analysis data suggested that the two blocks PCL and EPE in the copolymers tend to crystallize in separate phases. Furthermore, the presence of one component sensibly affects the crystallization of the other. In particular, since CL block crystallizes first, probably, the crystallization of the EO in EPE triblock is disturbed and eventually hampered, such as in the case of CUC4 copolymer.

#### 2.3.2. Wide-Angle X-Ray Diffraction (WAXD)

The phase attribution previously performed in the DSC data for PCL and EO crystalline blocks was verified with the WAXD analysis. Figure 6a,b shows WAXD patterns of CUC1–CUC4 and CUC5–CUC8 series, respectively, compared with pristine EPE triblock copolymer and PCL homopolymer.

EPE triblock copolymers have the same EO length and, consequently, the position of the two most significant diffraction peaks are at 2*θ* values of 19.1° and 23.2°, while the main diffractions of PCL are at 2*θ* values of 21.4° and 23.7°. The WAXD diffractograms of CUC copolymers basically are a superposition of those corresponding to the reference EPE and PCL, resulting in at least three diffraction peaks. The intensity of the sharp diffraction peak ata 2*θ* value of 19.1° (EPEs) decreased with the increasing of the CL block length. Besides, this peak almost disappears in CUC4 copolymer in agreement with the DSC data. This can be attributed to the fact that the melting temperature of EO block in CUC4 (Table 2) is very close to ambient, and thus it would be practically molten during WAXD study. On the other hand, the diffraction peak of CL block at a 2*θ* value of 23.7° moves to the high angle values of about 0.2°, which is in concordance to the T_m3_ increase (Table 2 and Table 3) with the increase in the CL block length. This shift may not be considered significant with respect to PCL homopolymer. So, no change in the unit cell dimensions and, hence, in crystal structure is evident. The diffraction peak of PCL at a 2*θ* value of 22° appears as a tail in the high angle side and in the CUC copolymers as a shoulder. This diffraction peak corresponds to the (111) plane of the orthorhombic PCL crystal, and its intensity decrease in relation to that of the (110) plane (2*θ* = 21.4°) indicates a change in its crystallographic texture, i.e., preferred orientation [23]. In conclusion, the results of WAXD analyses indicated that a micro-phase separation takes place in all the prepared copolymers.

#### 2.3.3. Morphology

Figure 7 and Figure 8 show typical polarized light optical micrographs (PLOM) of some copolymers morphologies crystallized at 35 °C after quenching from 90 °C. The difference in morphology between samples as a function of molecular weight of central EPE block and PCL is clearly observed.

EPE(F38) crystallized in a large Maltese cross extinction spherulite with some degree of disordered crystal structure (crystal with open texture), probably due to the amount of amorphous phase, developed in the crystallization condition used (Figure 7a). A Maltese cross texture with a smaller size and which was more ordered than EPE was found for PCL (micrograph not shown). On the other hand, the spherulite sizes of CUC copolymers were smaller than those of the corresponding EPE macro-initiator and equivalent to that of PCL but with different textures. Besides, it seems that the crystal size depends on the EPE molecular weight, as suggested in Figure 7b,c, and Atanase et al. showed equivalent morphologies in the crystallization of PCL having a triazole “defect” near the chain center [20].

CUC1 and CUC5 are copolymers with equivalent PCL/PEO ratio (47/42 and 45/44, respectively) but with ${\bar{M}}_{n_{SEC}}$ of ca. 9 kDa and 15 kDa, respectively (Table 1), which could be the factor responsible for the higher nucleation of CUC5. Moreover, the size of amorphous, incompatible PPO could also be the cause of the difference between them. Consequently, the length of the central block of CUC pentablock copolymers may have an important role in the crystal formation.

The size of spherulite did not show any relationship with the length of CL block in the CUC1–CUC4 series (Figure 7b and Figure 8). On the other hand, crystal sizes of the CUC5–CUC8 series were all equivalent to CUC5, shown in Figure 7c. This behavior suggests competitive factors and a systematic work is needed to individuate the key ones.

CUC spherulites showed two other features: one is the change in the interference colors without the use of a sensitive tint plate, and the other is the irregular “banding” texture. With the increase in the CL block length, the spherulites present increased yellow–orange–blue interference colors, an effect which was more significant in the CUC1–CUC4 series (Figure 8). The appearance of colors depends on the difference in the refractive index of the crystalline phase, inside the spherulite, and on the orientation of the crystals in relation to their radius. Regarding the texture of the band, it is optically caused by zero birefringence appearing as extinction. The lamellar torsion in the spherulite during its radial growth is one of the hypotheses formulated to explain the appearance of concentric bands.

The appearance of colors depends on the difference in the phase refraction index, on the crystal within the spherulite, and on the orientation of crystals with respect to the radius of the spherulite. Regarding the banding texture, it is optically caused by the zero birefringence appearing as extinction. Lamellar twist in spherulite during radial growth is one of the hypotheses formulated to explain the appearance of concentric bands [24].

### 2.4. Cytocompatibility Evaluation

In order to evaluate the biocompatibility of the polymeric materials, in vitro experiments were carried out with a 3T3/Balb clone A31 mouse embryo fibroblast cell line, and cell morphology and enzymatic activity were used to investigate the material toxicity [25]. Preliminary cytotoxicity tests were carried out on fluid extracts of CUC7 copolymer sample. Cells were incubated with the prepared extracts, undiluted or diluted 1:2 or 1:4 with complete growth media. Quantitative evaluation of metabolically active cells after exposure to polymer extracts was performed by incubation of cells with tetrazolium salt WST-1. In viable cells, mitochondrial dehydrogenase enzymatically converts WST-1 to formazan.

Cell viability was higher than 100% to the control material (glass), as shown in Figure 9, thus indicating that no cytotoxic compound is extracted from the polymeric material. Accordingly, this material can be defined as cytocompatible, confirming that the use of the enzymatic catalyst CALB is extremely convenient for the preparation of polymeric materials for biomedical applications, since it allows avoidance of the use of toxic metal catalysts.

## 3. Materials and Methods

### 3.1. Materials

All reagents were purchased from Sigma Aldrich Chemical Co., (St. Louis, MI, USA), Germany, unless otherwise stated. ε-Caprolactone (ε-CL, 98%) was dried over calcium hydride for 24 h at room temperature, and then distilled at reduced pressure (98 °C; 5 mm Hg) prior to polymerization. Pluronics F38 (EPE(F38)) and F68 (EPE(F68)) (kindly supplied by BASF Corporation, Mount Olive, NJ, USA) have the same EO content (*ca*. 80 wt%) and molecular weights of 4700 and 8400, respectively. EPEs were dried under vacuum at 60 °C for 48 h before use. Novozyme-435 (specified activity 7000 PLU/g according to the supplier) is the immobilized form of *Candida antarctica* lipase B (CALB) and was dried over anhydrous phosphorus pentoxide at 0.1 mm Hg for 48 h before use. Toluene was dried by refluxing over sodium metal, and then distilled under nitrogen atmosphere before use. All other chemicals were used as received.

### 3.2. Synthesis of PCL–EPE–PCL Pentablock Copolymers

The synthesis of pentablock copolymers PCL–EPE–PCL (henceforth called CUC) was carried out under rigorous anhydrous conditions, with a catalyst/ε-CL monomer ratio of 1/10 (*w*/*w*), solvent/monomer ratio of 2/1 (*v*/*w*), and different M/In ratios (Table 1). As an example, CALB (0.25 g) was added into a vial containing EPE(F38) (3.0 g, 0.64 mmol), ε-CL (2.6 g, 22.4 mmol), and 5 mL of toluene and stirred under nitrogen atmosphere at 70 °C for 4 h. The reaction was terminated by adding an excess of cold chloroform and stirring for 15 min. The insoluble enzyme was removed by filtration and washed with chloroform several times. The concentrated filtrate was precipitated in excess of diethyl ether. The recovered copolymers were further purified by re-precipitation in diethyl ether from chloroform solution. The precipitate was then washed with methanol to remove unreacted ε-CL monomer and EPE macro-initiator. The final product was collected as a fine white powder, which was dried under vacuum for 48 h.

### 3.3. Characterization

FT-IR spectra were recorded using a Perkin-Elmer FTIR spectrometer (Spectrum One) from 4000 cm^−1^ to 600 cm^−1^ at a resolution of 2 cm^−1^ by accumulating 16 scans. The samples were cast films from chloroform polymer solution on KBr plate. ^1^H-NMR spectra were acquired using a 200 MHz Varian Gemini nuclear resonance spectrometer in CDCl_3_, with CHCl_3_ hydrogen as the internal standard.

Size exclusion chromatography analyses (SEC) were performed at a flow rate of 1.0 mL/min by using a Jasco PU-1580 HPLC liquid chromatograph connected to Jasco 830-RI and Perkin-Elmer LC-75 spectrophotometer (λ = 260 nm) detectors, equipped with two Mixed-D PLgel columns (300 × 7.5 mm). Chloroform was used as the eluent and the calibration curve was established by using mono-dispersed polystyrene standards.

Differential scanning calorimetry (DSC) measurements were performed in a Mettler TA 4000 system consisting of the DSC-30 module and TA72 GraphWare software. Samples of 10–15 mg were weighed in a 40 μL aluminum pan and an empty pan was used as reference. DSC temperature and energy calibrations were carried out by using indium, lead, and zinc standards and indium standard, respectively. Measurements were performed under a nitrogen flow rate of 80 mL/min according to the following protocol:First heating scan from 25 °C to 150 °C at 10 °C/min;First cooling scan from 150 °C to –100 °C at 10 °C/min and 4 min of isotherm at –100 °C;Second heating scan from –100 °C to 100 °C at 10 °C/min.

Wide-angle X-ray diffraction patterns were performed at room temperature with a Kristalloflex 810 diffractometer (Siemens) using a Cu Kα (λ = 1.5406 Å) as the X-ray source. Scans were run in the high angle region 5° < 2*θ* < 40° at a scan rate of 0.016°/min and a dwell time of 1 s.

Polarized light optical microscopy (PLOM) observations were made by means of a Reichert-Jung Polyvar, equipped with a Mettler FP-52 hot-stage and Mettler FP-5 temperature controller. Samples were heated up to 90 °C and cooled quickly to the crystallization temperature at 35 °C and left for 30 min under nitrogen flow when pictures were taken.

### 3.4. Biological Tests

Experiments were performed following the standards indicated by ISO-10993-5 [26]. Cytotoxicity analyses were carried out on fluid extracts of the materials prepared by incubating 20 mg/mL of the polymeric sample and reference materials in complete cell culture medium (DMEM) for 48 h at 37 °C. Extracts were investigated either undiluted or at 1:2 and 1:4 dilution ratios with complete DMEM.

#### Quantitative Evaluation of Cytotoxicity

A subconfluent monolayer of 3T3/A31 fibroblasts was trypsinized in a 0.25% trypsin/1 mM EDTA solution. Cells were centrifuged at 1000 rpm for 5 min, redispersed in complete DMEM medium and counted. Appropriate dilutions were made in order to obtain 3 × 103 cells per 100 μL of medium, the final volume present in each well of flat–bottom 96-microwell plates. Plates were incubated at 37 °C in an atmosphere containing 5% CO_2_ for 24 h until 60–70% cell confluence was reached. The medium from each well was then removed and replaced with a fresh medium containing a different concentration of the material under investigation. Control cells were incubated with fresh growth medium, whereas wells containing only growth medium were used as a blank.

After 24 h of incubation with polymer extracts, cells were incubated with WST-1 cell proliferation reagent diluted at 1:10 (as indicated by the manufacturer) for 4 h at 37 °C in an atmosphere containing 5% CO_2_. Plates were then analyzed and the number of viable cells was evaluated using a Benchmark Bio-Rad Microplate Reader. Microplate absorbance measurements were performed at 450 nm, diminished by the blank absorbance at the same wavelength and by using 620 nm as a reference wavelength. All data were processed by using Microplate Manager III (Bio-Rad) and Igor Pro (Wave-metrics).

## 4. Conclusions

Amphiphilic pentablock copolymers with structure PCL–EPE–PCL were prepared by ring-opening polymerization of ε-caprolactone initiated by EPE dihydroxyl terminal groups and catalyzed by enzyme CALB. The polymers were obtained with yields between 70 and 85% and characterized by FT-IR, ^1^H-NMR, SEC, WAXD, DSC, PLOM, and by cytotoxicity tests. The enzymatic ROP showed dependence on reaction time and controlled polymerization features.

CUC copolymer crystal organizations were stable for the thermal history performed in this study and presented up to three endothermic transitions in DSC analysis. The melting transition at a higher temperature, attributed to CL block, showed a linear relationship with its weight fraction in the copolymer. This result was also confirmed by WAXD analysis. Besides, CL block has a different crystallographic pattern in the copolymers, as indicated by the peak corresponding to the (111) plane. PLOM observations suggested competitive factors in the crystal formation at 35 °C, which very likely can be associated with the PO central block length and CUC molecular weight. Two interesting spherulite features were observed in the polymer crystals: interference colors and banding. Cytotoxicity tests indicated that CUC copolymers are nontoxic, confirming the suitability of the applied enzymatic catalysis for the preparation of polyester/polyether copolymers for biomedical applications that, at good right, may be respondent to their use in biomedical/pharmaceutical applications and, therefore, further studies are being carried out in this direction.

## Data Availability

Not applicable.

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
