# Peer review of "Amphiphilic Pentablock Copolymers Prepared from Pluronic and ε-Caprolactone by Enzymatic Ring Opening Polymerization"

_ijms, 2022, doi:10.3390/ijms23031390_

Round 1
Reviewer 1 Report
Please see the attached file.

Author Response
"Please see the attachment."

Reviewer 2 Report
The paper submitted by El-Fattah et al. deals with the synthesis and characterization of two series of copolymers based on Pluronic and PCL. These copolymers were obtained in the absence of any toxic catalysts which make them suitable for biomedical applications.
The manuscript is clear, well written and and conclusions are supported by the results. However, some minor corrections are needed before publication:
- in the abstract section, please use do not use abbreviations without providing firstly the full name
- actually the term "polydispersity index PDI" is no longer accepted and should be replace with "Dispersity Đ"
- in the section about the solid state characterization, please use the following reference: https://doi.org/10.1016/j.polymer.2011.05.017
- in fig 9, please add the data for positive and negative control.
Author Response
"Please see the attachment."
